# Ibrutinib Could Suppress CA-125 in Ovarian Cancer: A Hypothesis

**Julian Matthias Metzler** [1,*] **, Daniel Fink** [2] **and Patrick Imesch** [1]

1   Department of Gynecology, University Hospital Zurich, Frauenklinikstrasse 10, 8091 Zurich, Switzerland;
    Patrick.Imesch@usz.ch
2   Faculty of Medicine, University of Zurich, Pestalozzistrasse 3/5, 8091 Zurich, Switzerland; D.Fink@hin.ch
*   Correspondence: Julian.Metzler@usz.ch; Tel.: +41-442551111

**Abstract:** Ibrutinib is a small-molecule inhibitor of Bruton's tyrosine kinase, an enzyme central in B cell development. It is indicated as a therapy for certain hematological diseases such as chronic lymphocytic leukemia (CLL), but also exerts off-target effects on several receptors and kinases. In this paper, we hypothesize that ibrutinib may suppress the tumor marker CA-125 in ovarian cancer. The hypothesis is based on an observation of CA-125 normalization in a patient with low-grade serous ovarian cancer who received ibrutinib for concurrent CLL. We propose a mechanistic model explaining this possible drug effect as a foundation for further research.

**Keywords:** ibrutinib; Bruton's tyrosine kinase; Btk; kinase inhibitor; ovarian cancer; gynecological oncology; solid tumors; CA-125

## 1. Introduction

Ibrutinib is a small-molecule inhibitor of Bruton's tyrosine kinase (Btk), an essential enzyme in B cell differentiation. Indications for oral ibrutinib include certain hematological diseases such as chronic lymphocytic leukemia (CLL), mantle cell lymphoma (MCL), or marginal zone lymphoma (MZL). Apart from its affinity to Btk, ibrutinib exerts off-target effects on several receptors and kinases such as the epidermal growth factor receptors (EGFRs 1-4), interleukin-2-inducible T-cell kinase (ITK), or Janus kinase 3 (JAK3). Influencing these additional molecular targets, ibrutinib has been shown to be effective not only in hematological malignancies but also in solid tumors. Nevertheless, evidence of ibrutinib affecting ovarian cancer is scarce and limited to preclinical trials [1–5].

CA-125 is a heavily glycosylated protein and a tumor-associated antigen. It is the epitope of MUC16, a high-molecular-weight transmembrane mucine, occurring in the pleura and the peritoneum as well as in the female reproductive tract epithelia, the ocular surface, and the respiratory tract [6]. Physiologically functioning as a protecting and lubricating agent, the altered expression and glycosylation of MUC16 have been identified in ovarian carcinoma [6] and MUC16 downregulation has been linked with increased cisplatin sensitivity [7]. Even though CA-125 is the most widely used tumor marker in ovarian cancer, elevated serum levels have a limited specificity, allowing for a broad range of benign differential diagnosis. This serum increase in non-malignant or inflammatory conditions is known to be mediated by a series of cytokines such as IL-1β, IL-6, IL-8, IL-17, TNFα, and IFNγ [6].

Nonetheless, there are insufficient data about the regulation and expression of CA-125. Regarding malignancy, recent research has demonstrated an important role of mesothelial cells in MUC16 production and has suggested malignant ascites as a strong modulator of its expression [6]. Upregulation of MUC16 has been shown to be regulated via the KRAS/ERK axis [8]. A thorough literature search revealed no information about ibrutinib's influence on its expression or regulation.

As a clinical introduction, we first present the case of a patient with ovarian cancer, whose CA-125 levels normalized after the initiation of ibrutinib therapy for concurrent CLL. To the best of our knowledge, this synchronism has not been described in the literature until now, providing evidence for our hypothesis of a possible, formerly unknown drug-effect. Secondly, the hypothesis is presented and evaluated considering the available literature.

## 2. Case Report

A 61-year-old woman was referred to our department in 2010 for further treatment of an enlarged ovary, which had been an incidental finding in a follow-up CT scan for CLL. The CLL had previously been treated with chlorambucil and prednisone. Secondary diagnosis included arterial hypertension and dyslipidemia. Upon referral, transvaginal sonography revealed an ovarian tumor of 57 × 30 mm containing solid and cystic components. Blood serum analysis was negative for CA-125. Surgical treatment was performed as follows: hysterectomy, adnexectomy, pelvic and para-aortal lymphadenectomy, incidental appendectomy, and radical omentectomy. Histological workup revealed a papillary serous borderline tumor with transition to a low-grade ovarian cancer, stage FIGO (Fédération internationale de gynécologie obstétrique) IIIb. Postoperatively, an adjuvant platinum-based therapy (6 cycles of carboplatin/paclitaxel) was administered as standard of care.

In 2011, the patient was diagnosed with sarcoidosis. Systemic steroids were administered from December 2011 until April 2012, resulting in total remission. In 2013, the patient suffered her first recurrence of CLL. The treatment regimen included 2 cycles of ribomustin and rituximab.

In 2014, rising levels of CA-125 marked recurrent ovarian cancer, and a re-laparotomy for tumor-debulking and partial colon resection with end-to-end anastomosis was performed in October 2014. Second-line chemotherapy contained 6 cycles of carboplatin and gemcitabine until March 2015. In July 2016, sixteen months after second-line chemotherapy, elevated CA-125 levels were recorded, without further clinical or radiological findings, allowing for expectant management of the asymptomatic patient (Figure 1). Gynecological follow-up visits were continued every 3 months.

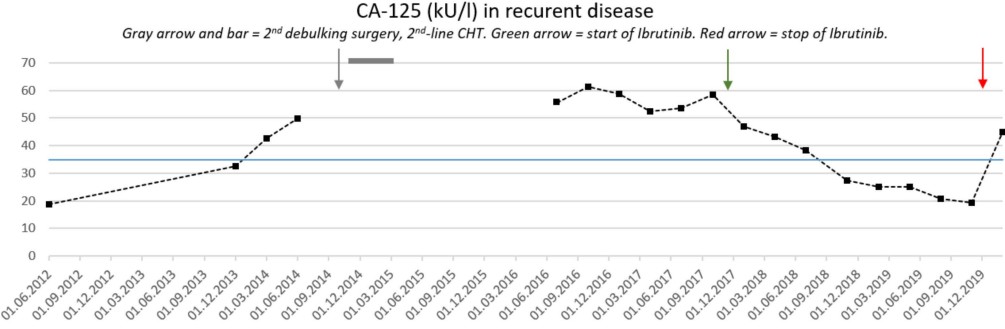

**Figure 1.** Development of CA-125 levels in recurrent disease. An increase was seen during the course of the disease, and debulking surgery with second-line chemotherapy was performed (grey arrow and grey bar). CA-125 monitoring was not continued postoperatively until July 2016, when elevated levels suggested recurrent (subclinical) disease. During expectant management, CA-125 dropped sharply after the initiation of ibrutinib therapy for a different indication (green arrow). After termination of ibrutinib intake (red arrow), a marked increase was seen.

A second recurrence of CLL was diagnosed in May 2017. Repeated therapy with ribomustin and rituximab showed minimal response. Therefore, an oral therapy with ibrutinib (140 mg, alternating 2×/3× daily) was established in November 2017, which allowed for hematological disease control.

Following the initiation of ibrutinib, the CA-125 levels started to decrease continuously, with normalization after 12 months of therapy. During the continued therapy, the patient had no laboratory, radiological, or clinical signs of recurring ovarian cancer.

We initiated genetic testing of the cancer sampled in the second surgery via Foundation One CDx (Foundation Medicine Inc., Cambridge, MA, USA; Hoffmann La Roche AG, Basel, Switzerland). The molecular profile showed the genomic signature of a microsatellite-stable carcinoma with a loss-of-heterozygosity score of 0.0% and a low tumor mutational burden (3 Muts/Mb). Gene alterations were found in ARID1A and MUTYH, as well as an activating oncogenic KRAS G12V mutation. None of the mutations offered any therapeutic consequences, as confirmed by our molecular tumor board.

In January 2020, ibrutinib was stopped by the treating hematologist after 26 months of therapy due to laboratory signs of hepatitis. The patient was hospitalized due to a deterioration of general condition shortly thereafter. The signs and symptoms were interpreted in the context of relapsing sarcoidosis and treated accordingly. In February 2020, six weeks after discontinuation of ibrutinib, an elevated CA-125 level was registered, without any clinically or radiologically apparent tumor.

## 3. Hypothesis

In our case report, we describe a patient with low-grade ovarian cancer experiencing a prolonged normalization in CA-125 under concurrent ibrutinib treatment. Although not necessarily causal, the timely fashion of the protein decline and subsequently incline after termination of treatment is staggering.

From these observations, the next step was to formulate a bio-molecular hypothesis that could mechanistically explain a decrease in CA-125 after ibrutinib intake. We proposed the following model, as depicted in Figure 2. Ibrutinib inhibits the phosphorylation of EGFR (1–4) as well as Src [9] and the FGFR in mesenchymal cells [10]. This blocks downstream effectors and stops EGFR/Src from activating RAS [11]. Subsequently, the oncogenic RAS pathway, which has been shown to encourage MUC16 upregulation via RAF, MEK, and ERK [8], is not activated. Finally, MUC16 production and CA-125 shedding diminish. MEK inhibitors affect proteins further downstream the pathway and have been described to normalize CA-125 levels [12].

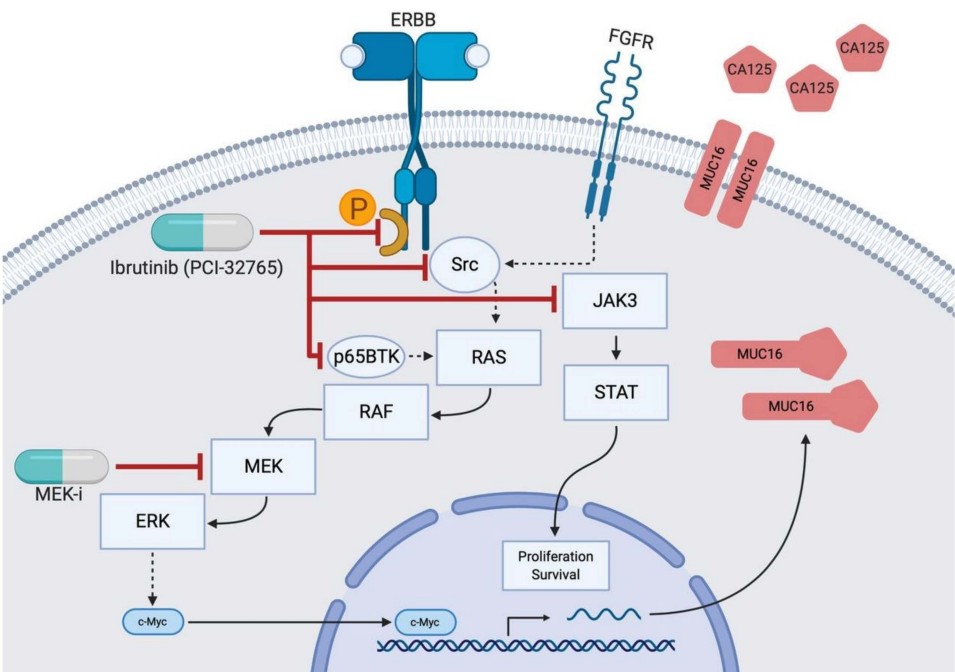

**Figure 2.** Proposed mechanism of ibrutinib-mediated suppression of CA-125 production along the RAS–RAF–MEK–ERK pathway including additional inhibitory sites (created with Biorender.com).

## 4. Discussion

There are several limitations to our hypothesis. It is important to stress that the observed correlation does not imply causation. Nevertheless, the absence of other concomitant oncological medication during the time of the CA-125 decrease in our case report suggests at least a partial effect of ibrutinib.

Randomly elevated CA-125 levels due to sarcoidosis have sporadically been described in case reports [13]. These reports are rare, implying a correlation due to the absence of differential diagnosis or due to involvement of serosal surfaces. In our patient, sarcoidosis was primarily cutaneous and pulmonal, without pleural or peritoneal effusions, and CA-125 was not elevated during the initial diagnosis of sarcoidosis in 2011. The tumor marker has not since been established for monitoring sarcoidosis activity. Concerning the observed decline of CA-125, spontaneous regression is extremely rare in epithelial ovarian cancer. A PubMed search (1 May, 2020) revealed only one case report of spontaneous regression in recurrent disease after radiation of a single nodal metastasis [14]. As spontaneous regression and sarcoidosis fail to explain the serum marker fluctuation, the kinase inhibitor's possible role needs to be investigated.

Few in vitro experiments have studied ibrutinib's effect on ovarian cancer. Papillary serous cells, such as in the present case, displayed drug response in ex vivo drug sensitivity testing [15]. Ovarian cancer stem cells have been shown to express Btk, with ibrutinib diminishing their self-renewal capacities [5].

Regarding other kinase inhibitors, recent case reports describe disease response and declining CA-125 after therapy with trametinib, a MEK inhibitor affecting the MAPK/ERK pathway [12]. Furthermore, we found a case report about a patient harboring the same KRAS mutation who experienced an impressive clinical response after treatment with binimetinib, another MEK inhibitor [16]. Animal studies found overexpression of p65BTK, an isoform of Btk, in KRAS-mutant cell lines. Btk inhibitors proving effective against cell viability in these experiments offers another molecular explanation for our findings [17].

CA-125 expression shows a high correlation with serous subtype borderline and malignant ovarian tumors [18], and we conclude that the varying levels of CA-125 in our patient were a direct result of ibrutinib's effect on subclinical ovarian cancer. The absence of a radiologically or clinically detectable tumor during this period is consistent with this conclusion, as low tumor volumes typically elevate serum levels before they become clinically evident. The observed decrease in CA-125 during ibrutinib intake can be explained by a cytotoxic effect, a metabolic effect (suppression of CA-125 production in sub-apoptotic serum levels), or a combination of these.

Further research is needed to test this hypothesis in low- and high-grade ovarian cancer, and we encourage the conduction of in vitro experiments exploiting the underlying pathways for tumor suppression. In this context, our work can be seen as an additional puzzle piece in this widely unexplored area, bridging the gap toward more clinically relevant research.

## 5. Conclusions

To the best of our knowledge, the described case is the first published report of ibrutinib possibly suppressing CA-125 levels, implying a surrogate ovarian cancer suppression. The temporal context of starting ibrutinib intake and the subsequent decrease in tumor marker levels further support the preclinical evidence that this kinase inhibitor's multiple anti-neoplastic effects can be used against ovarian cancer. We proposed a possible mechanism of action leading to CA-125 suppression. Future research should focus on ibrutinib's capabilities in CA-125 suppression and the underlying processes involved. This may ultimately lead to an extended clinical use of this drug.

**Author Contributions:** Conceptualization, J.M.M. and D.F.; writing—original draft preparation, J.M.M.; writing—review and editing, P.I.; visualizations, J.M.M.; validation, P.I.; supervision, P.I. and D.F.; project administration, all authors. All authors have read and agreed to the published version of the manuscript.

**Funding:** This research received no external funding.

**Institutional Review Board Statement:** According to Swiss law, this is not a research project under the Swiss Human Research Act (Humanforschungsgesetz, HFG), and therefore, no authorization is required. Written informed consent was obtained from the patient for publication of the case report and the accompanying images.

**Informed Consent Statement:** A copy of the written consent is available for review by the Editor-in-Chief of this journal upon request.

**Data Availability Statement:** The datasets used and/or analyzed during the current study are available from the corresponding author on reasonable request, to the limit where individual privacy could be compromised.

**Acknowledgments:** Thanks to Carla Trachsel for proofreading the article.

**Conflicts of Interest:** The authors declare no conflict of interest.

## Abbreviations

| | |
|---|---|
| Btk | Bruton's tyrosine kinase |
| CLL | Chronic lymphocytic leukemia |
| EGFR | Epidermal growth factor receptor |
| FIGO | Fédération internationale de gynécologie obstétrique |
| FGFR | Fibroblast growth factor receptor |
| HPMC | Human peritoneal mesothelial cell |
| IFN | Interferon |
| IL | Interleukin |
| ITK | interleukin-2-inducible T-cell kinase |
| MEK(-i) | mitogen-activated protein kinase (-inhibitor) |
| MZL | Marginal zone lymphoma |

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
