# Peer review of "Ibrutinib Could Suppress CA-125 in Ovarian Cancer: A Hypothesis"

_applsci, doi:10.3390/app11010222_

Round 1

Reviewer 1 Report

Summary

The manuscript titled ‘Ibrutinib could suppress CA-125 in ovarian cancer: A Hypothesis’ describes an observation based on a case study. During the treatment of a patient for chronic lymphocytic leukemia, the authors observed the decrease of a biomarker CA-125, whose levels are associated with different pathologies including ovarian cancer. During the duration of ibrutinib treatment, there was a decrease in the levels of CA-125 followed by an increase after treatment with ibrutinib. The authors provide a hypothesis to explain this decrease of CA-125 in response to ibrutinib.

Strengths

The article is well-written. The authors have explained the chronological sequence of events of the patient and provided appropriate details of their case study. Furthermore, the authors have admitted some limitations to their hypothesis as well.

Weaknesses

  1. The main weakness is that the title and the manuscript refer to the suppression of CA-125 by ibrutinib in the context of ovarian cancer. Based on the author’s descriptions, the patient was diagnosed with recurrent ovarian cancer in 2014 that was marked with an increase in CA-125. After performing a second-line of chemotherapy, the authors state that ‘CA-125-levels in the asymptomatic patient continued to rise slowly without further clinical or radiological findings’. This statement implies that the elevation of CA-125 levels was independent of ovarian cancer. Furthermore, after discontinuing ibrutinib treatment, there was an increase in CA-125, but there is no mention of the status of ovarian cancer. Hence it is not clear if the increase in CA-125 levels before ibrutinib treatment, and subsequent decrease followed by increase is CA-125 is related to ovarian cancer. The authors are requested to address this aspect since it is known that the levels of CA-125 can increase in the absence of ovarian cancer.
  2. Another weakness of the manuscript is that the entire observation and hypothesis is based on one patient. It is understandable that it is difficult to have an adequate number of patients to observe in such scenarios, which is a limitation that is beyond the control of the authors. However, the authors should be able to test the ability of ibrutinib to inhibit/suppress CA-125 using other methods like in vitro studies. This would provide more support to the study and strengthen it.
  3. In the conclusion, the authors propose that future studies should focus on how ibrutinib is able to decrease CA-125 levels, but it is not clear what this decrease means from a biological standpoint as CA-125 is used as a biomarker, but is not known to cause a pathology. It would be better if the authors could elaborate on whether they think that suppression of CA-125 by ibrutinib results in or is indicative of better outcomes to help the reader appreciate the importance of reducing CA-125 levels.   

Author Response

Dear Reviewer,

please find our reply in the attached word file.

Kind regards,

Julian Metzler

Reviewer 2 Report

Metzler and colleagues tried to explore the relationship between Ibrutinib treatment and CA-125 expression in ovarian cancer.

The study is novel, but there are some issues to be addressed.

  1. What is the potential clinical benefit of this study?
  2. Were there detectable ovarian tumors during the Ibrutinib treatment? If no, where is the CA-125 from? If yes, was the CA-125 decrease due to the tumor control?
  3. It is easy to test whether Ibrutinib can suppress CA-125 expression in vitro. The authors need to test it by using human ovarian tumor cell lines.
  4. The date labeled in Figure 1 does not matches what is described in the text.

Author Response

(The authors gave the same response as above.)

Round 2

Reviewer 2 Report

Most of my questions are addressed. Agree to publish.